# How Many People Experience Unsafe Medical Care in Thailand, and How Much Does It Cost under Universal Coverage Scheme?

**DOI:** 10.3390/healthcare11081121

**Published:** 2023-04-13

**Authors:** Vilawan Luankongsomchit, Chulathip Boonma, Budsadee Soboon, Papada Ranron, Wanrudee Isaranuwatchai, Nopphadol Pimsarn, Piyawan Limpanyalert, Ake-Chitra Sukkul, Netnapa Panmon, Yot Teerawattananon

**Affiliations:** 1Health Intervention and Technology Assessment Program, Nonthaburi 11000, Thailand; 2Healthcare Accreditation Institute (Public Organization), Nonthaburi 11000, Thailand

**Keywords:** adverse events, medical harm, patient safety, unsafe care, inpatient, Universal Coverage scheme, Thailand

## Abstract

Adverse events and medical harm comprise major health concerns for people all over the world, including Thailand. The prevalence and burden of medical harm must always be monitored, and a voluntary database should not be used to represent national value. The purpose of this study is to estimate the national prevalence and economic impact of medical harm in Thailand using routine administrative data from the inpatient department electronic claim database under the Universal Coverage scheme from 2016 to 2020. Our findings show that there are approximately 400,000 visits with potentially unsafe medical care per year (or 7% of all inpatient visits under the Universal Coverage scheme). The annual cost of medical harm is estimated to be approximately USD 278 million (approximately THB 9.6 billion), with an average of 3.5 million bed-days per year. This evidence can be used to raise safety awareness and support medical harm prevention policies. Future work should focus on improving medical harm surveillance using better data quality and more comprehensive data on medical harm.

## 1. Introduction

The World Health Organization (WHO) defines an adverse event as an incident that causes injury and/or any deleterious effect to a patient because of decisions made or actions taken during the provision of healthcare [1]. These adverse events can be preventable or unpreventable. Preventable events consist of medical harm that can be avoided with standard care.

Medical harm has emerged as a major issue in healthcare quality fields over the past few years, which have seen the publication of “To Err is Human: Building a Safer Health System” [2] and “Crossing the Quality Chasm: A New Health System” [3] from the United States. These reports revealed that medical harm was regularly found in healthcare services and had adverse impacts on patients. Additionally, the solution to this problem would be to employ comprehensive approaches to improve patient safety, which would require the collaboration of all stakeholders.

According to the Organization for Economic Cooperation and Development (OECD), 1 in every 10 patients suffers harm at the point of care [4]. Furthermore, more than 10% of total medical expenditure in developed countries is spent on correcting preventable harm, such as treating postoperative sepsis, correcting medication errors, or employing inappropriate emergency services [5]. Because of the widespread impact of medical harm, WHO member states adopted the 2021–2030 Global Patient Safety Action Plan to reduce avoidable harm caused by substandard care [6].

Thailand, an upper–middle-income country in Southeast Asia, has a national policy known as “Patient and Personnel (2P) Safety”, which aims to improve the quality and safety of healthcare services affecting both patients and personnel. The 2P Safety policy was adopted in 2016 by the Thai Ministry of Public Health (MoPH) and 15 other health organizations and professional councils, including the Healthcare Accreditation Institute (Public Organization), or HAI, which serves as the national agency responsible for this issue. Before the national policy announcement, Thailand conducted a self-assessment of its patient safety and discovered several challenges, such as the lack of a national monitoring and of an evaluation system to track long-term progress [7]. As a result, Thailand created the National Reporting and Learning System (NRLS), which has been used as the country’s medical harm database since 2017. However, the database still has limitations, such as not covering all hospitals (covering only 58% of both public and private hospitals as of 2022), relying on a voluntary reporting mechanism, and being unable to link the reported cases with information on healthcare costs for those specific cases [8]. Therefore, it was not feasible to reflect the national prevalence and economic burden of these adverse events, which suggests that Thailand lacks the proper evidence to influence and raise patient safety awareness among policymakers and citizens.

In many countries, such as the United States, Canada, and Australia, routine hospital administrative data have been chosen as reliable sources for determining the prevalence and economic burden of adverse events [9,10]. These countries have developed patient safety indicators (PSIs) by using ICD-10 codes to identify the prevalence of adverse events in hospitals (e.g., pressure ulceration and patients falling). In Thailand, there is a comparable database, known as the IPD e-claim (inpatient department electronic claim) system, under the Universal Coverage scheme (UCS). This inpatient database contains individual health information and reimbursement costs for 70% of the Thai population, including newborn, teenage, adult, and elderly individuals who are not covered by the Social Security Scheme (SSS) for formal private sector employees, or the Civil Servant Medical Benefit Scheme (CSBMS) [11] for government officials and their dependents. Thus, this study aims to use these routine administrative data to estimate the national prevalence and economic impact of inpatient medical harm.

## 2. Materials and Methods

### 2.1. Study Design and Data Source

This study is a retrospective secondary data analysis using data from 2016 to 2020 from the IPD e-claim database of the National Health Security Office (NHSO). The data include information on hospitalized patients, defined as a patient who stayed in a hospital for more than six hours. These time periods (2016 to 2020) were chosen due to their data accessibility. Before retrieving data from the database, all data were deidentified, encrypted, and anonymized. There was no risk to the patients or of identifying them; therefore, no consent was needed for this study.

The variables considered in this study consisted of hospital code, sex, age, primary diagnosis code (ICD-10 codes), secondary diagnosis code (ICD-10 codes), summary cost, and length of stay. The summary cost variable was the actual reimbursement cost that was paid by the NHSO to the hospital for each patient’s visit. It included the costs of health services, general drugs, disease prevention services, laboratory analysis, artificial organs, and medical devices (which were reimbursed through an electronic claim system and did not include specialized treatments such as high-cost drugs and procedures).

### 2.2. Identifying ICD-10 Codes Related to Medical Harm

We used the Canadian PSIs or ICD-10 codes from the Southern (2017) study [10] to identify visits potentially related to medical harm. Unlike in other studies, these codes were developed using the national discharge database rather than through a literature review. Moreover, all codes were additionally reviewed and accepted by experts via the Delphi panel process to ensure their suitability for PSIs.

We finalized the ICD-10 codes with the approval of clinicians and HAI staff members who are medical harm specialists in Thailand to ensure their proper use. Finally, we obtained 58 PSI codes for the data analysis step (Appendix A). The accuracy of PSIs in identifying real adverse events is not perfect. The true-positive rate is affected by the hospital setting and the type of event; for example, the positive predictive value (PPV) for decubitus ulcers is approximately 51%, whereas the PPV for postoperative respiratory failure in the same database is 14% [12]. In this study, we used ideal conditions, in which PSI accuracy equaled one, and did not reduce the number of adverse events in Thailand by the true-positive rate.

### 2.3. Analysis

Descriptive statistics were used to estimate the number of medical harm events and associated burdens for each year (including the sum, median, average, interquartile range, and standard deviation). The prevalence of medical harm under the UCS was calculated by dividing the total number of medical harm incidents by the total number of inpatient visits in each year. Medical harm can occur more than once in a single visit (≥2 events per visit). To avoid the duplication of impact counts, estimated impacts (costs and lengths of stay) were counted by visit rather than by event (i.e., visits were assumed to be independent of one another).

All costs were converted to a base year in 2019 using the inpatient category of Thailand’s consumer price index (CPI) [13], to equalize costs to the same year (Appendix A). After this, the currency was changed from THB to USD using the exchange rate on 9 December 2022 (THB 34.64 = USD 1) [14]. All of the analyses in this study were performed using STATA software, version 17. Simple regression models were used to assess the differences in the number of medical harm events and cost among age groups and hospital types, where *p*-values of less than 0.05 were considered to be statistically significant.

## 3. Results

There were approximately six million total hospital visits per year available in the database. Of these, 400,000 visits annually (or 7% of the inpatient visits) involved unsafe medical care while hospitalized. This trend was quite stable over time (Table 1) (Appendix A).

The annual cost of medical harm in inpatient care was estimated to be approximately USD 278 million (approximately THB 9.632 billion), with an average of 3.5 million bed-days per year (Table 1). When the results were compared among age groups, medical harm was more frequently found in elderly individuals (≥60 years old) and in other hospital types (for example, medical schools and health centers) (Table 2). In terms of costs, the elderly and central hospital groups showed the highest burden of medical harm (Table 3 and Table 4). The number of medical harm events and related costs were significantly different among age groups and hospital types over time.

Overall, the top ten causes of unsafe inpatient care were sepsis, bacterial infection, decubitus ulcers, procedure complications, perineal laceration, postprocedural disorders, and complications with labor and delivery (Table 5).

## 4. Discussion

The estimated prevalence of medical harm in Thailand was approximately 7% among hospitalized patients under the Universal Coverage scheme. Our result was quite similar to that of the global report, which was approximately 6% [15]. However, fewer than 12.7% of adverse events have been reported in low- and middle-income countries [16]. This difference could have been caused by data limitations at the time, which included a lack of high-quality data and no coverage of Asian countries.

This study discovered a higher prevalence of medical harm in elderly patients, which is consistent with studies conducted in the United States [17,18]. The highest prevalence occurring in other types of hospital (such as medical university hospitals and health centers) can be attributed to the fact that admitted patients in those facilities are typically more severely ill than other patients and are referred from other hospitals. As such, they are more likely to have hospital-acquired infections than other patients.

Regarding its macroeconomic impact, our study indicates that medical harm has a significant impact on approximately 5.5% of the UCS budget [19]. This estimate significantly differs from that reported by the Organization for Economic Cooperation and Development (OECD), which estimated that medical harm costs approximately 15% of the total medical expenditure [20]. This difference may result from using a different method to estimate the burden (e.g., the difference in cost components or prevalence rates) and the fact that our current analysis included only one public health insurance scheme. Furthermore, the OECD study relied on evidence from extensive research across European countries, resulting in a wide range of economic impacts on 1.3% to 32% of the total health budget.

The number of medical harm events and related costs was found to be statistically different across age groups and hospital types. Given the large sample size, these findings do not necessarily imply that the number of medical harm events increased with age or were more prevalent in certain types of hospital. The results highlight that age and hospital type should be considered when study medical harm in the future, and those involved in the planning of policies to prevent medical harm may want to consider age and hospital type (such as tailoring an intervention specifically for certain age groups) [21].

Regarding the results of this study, infections, ulcers, and complications were the leading causes of medical harm in Thai inpatient populations, which is consistent with findings from other studies [15,16,18]. Infections, especially healthcare-associated infections (HAIs), can be caused by inadequate hand hygiene, the improper use of medical devices, or environmental troubles in healthcare settings (e.g., contaminated surfaces). Ulcers can be caused by long-term pressure from immobility, and complications can arise from a variety of factors, for example, medication errors or inadequate patient monitoring [2,22]. Many patient safety interventions were compiled and summarized in this systematic review to reduce those types of medical harm, such as electronic system use, checklist use, behavioral change interventions, process interventions, managerial and organizational interventions, patient-centered interventions, and patient and staff education [23]. Additionally, it is essential to choose the appropriate interventions to control and reduce medical harm. All patient safety interventions have advantages and disadvantages that policymakers and users must consider when making decisions.

Understanding the prevalence and impact of medical harm can help inform decisions about prioritizing funding to reduce the burden. These findings should encourage policymakers to invest in interventions, including surveillance systems and patient safety interventions, to reduce medical harm and its impacts. Policymakers, particularly at the national and organizational levels, should learn from the mandatory reporting system while also encouraging voluntary reporting at the same time, to build a data support system for informing safety policy [2]. Moreover, they should create a culture and environment of safety by raising safety standards, enhancing leadership, and increasing the knowledge of personnel and patients to ensure that the system can continue long-term [2].

This study demonstrates the possibility of using routine hospital data to monitor medical harm in a middle-income country; nevertheless, there are several limitations. First, the study did not correct the accuracy of the clinical diagnosis of admitted patients. The sensitivity and specificity of diagnoses may be context-specific across hospital types, patient groups, and settings [10,24]. While this issue is beyond the scope of our current study, we urge authors of future research to address this important point. Second, diagnosis timing was not indicated in the hospital data; for example, some events may have occurred prior to hospital admission. As a result, it is likely that our reported prevalence and economic impact numbers are overestimated. Canada, the United States, and Australia have developed a diagnosis timing indicator known as “present on admission” (POA) to help correct this limitation; this approach should be recommended for the future development of hospital data in Thailand and beyond. It is important to note that the reliability of POA is dependent on the quality of medical records and the coder’s appraisal skill [24,25,26]. This information means that the training of hospital staff who are responsible for electronic medical record coding is also crucial and should be part of the 2P Safety policy in Thailand. The assumption that the PSI accuracy equaled one may have led to further overestimation of the reported prevalence and economic impact of adverse events.

Third, the costs reported as the economic burden included the treatment costs of the primary disease. For example, the cost of treating surgical wound infection may include the costs of surgical treatment, unless the patients were admitted for surgical wound infection as a primary cause. Unfortunately, the e-claim database cannot distinguish patients in these two groups; thus, we cannot differentiate the first from the second. This situation may have caused further overestimation of the economic impact of adverse events. Fourth, the events in this study did not include medical harm in emergency wards or outpatient departments for UCS patients. Additionally, they did not include medical harm that occurred in patients under the other two health insurance schemes: the Social Security Scheme (SSS) and the Civil Servant Medical Benefit Scheme (CSMBS). Minor medical harm events, such as taking the wrong medication or overdosing without serious symptoms, or near-miss events, such as correctable wrong prescriptions, were not part of the reports used in this study, as the database did not comprehensively capture these minor cases. Last, economic burden focused only on direct medical care costs, excluding indirect or intangible costs such as patient or family distress, posthospitalization expenditure, staff burn-out, and the costs of medical lawsuits. This exclusion of other costs could have led to substantial underestimation of the economic burden of adverse events, which may be balanced out based on the previous assumptions, which may have overestimated the economic burden.

Future work could build on the lessons learned from this study and explore ways to estimate and monitor medical harm in the long run. For example, future research could compare these findings with the voluntary reporting system, i.e., NRLS, to understand the similarities and discrepancies between the two databases, in order to improve the accuracy and completeness of medical harm reports in Thailand. Moreover, qualitative information, such as the perception and awareness of stakeholders on this topic, will be needed to support the movement toward patient harm policies. These include different concepts of healthcare quality among different groups of stakeholders. The qualitative component of medical harm policy is crucial, because healthcare services are regarded as involving both affective and emotional labor that is intended to produce an emotional response in patients and, at the same time, require feelings and expression in healthcare workers to fulfil the emotional requirements of the job [27,28].

## 5. Conclusions

In conclusion, this study aims to illustrate the feasibility of assessing the prevalence and potential economic impacts of medical harm in Thailand using a routine administrative database. Despite the data limitations, the study findings can be used to raise awareness about and support policies that prevent medical harm. Future work should focus on improving medical harm surveillance through better data quality and comprehensive data on all adverse events, in all types of patient, and from all public health insurance policies. This study attempts to prioritize work and monitor the progress and success of the 2P Safety policy in Thailand.

## Figures and Tables

**Table 1 healthcare-11-01121-t001:** Number of Thai inpatient medical harm events between 2016 and 2020.

Year	Number of Inpatient Visits (Visits)	Number of Medical Harm Events (Visits)	Prevalence of Medical Harm (%)				Estimated Burden of Medical Harm Events		
				Median Cost (USD)	Interquartile Range (USD)	Total Cost (Million USD)	Median Length of Stay (Days)	Interquartile Range (USD)	Total Length of Stay (Days)
2016	6,056,500	424,411	7.01	271.74	625.23	261.60	5	9	3,647,006
2017	6,006,660	422,869	7.04	288.97	623.85	263.58	5	8	3,681,199
2018	6,264,661	427,341	6.82	299.11	632.85	270.26	5	8	3,690,152
2019	6,366,463	439,386	6.90	312.21	645.96	283.63	5	8	3,761,566
2020	5,758,165	458,034	7.95	343.69	691.14	311.29	5	8	2,836,292
Average	6,090,490	434,428	7.13	303.14	643.81	278.07	5	8	3,523,243

**Table 2 healthcare-11-01121-t002:** Number of medical harm events in each group.

	Year 2016		Year 2017		Year 2018		Year 2019		Year 2020	
	Number of IPDs	Number of Medical Harm Events	Number of IPDs	Number of Medical Harm Events	Number of IPDs	Number of Medical Harm Events	Number of IPDs	Number of Medical Harm Events	Number of IPDs	Number of Medical Harm Events
Total visits	6,056,500	424,411	6,006,660	422,869	6,264,661	427,341	6,366,463	439,386	5,758,165	458,034
		(7.01%)		(7.04%)		(6.82%)		(6.90%)		(7.95%)
Age group *										
0–14	1,677,053	54,269	1,575,436	50,746	1,691,317	51,857	1,646,175	48,726	1,283,621	46,613
	(27.70%)	(3.24%)	(26.23%)	(3.22%)	(27.00%)	(3.07%)	(25.86%)	(2.96%)	(22.29%)	(3.63%)
15–59	2,483,391	162,461	2,483,816	160,335	2,530,714	157,192	2,562,273	158,014	2,404,989	161,855
	(41.00%)	(6.54%)	(41.35%)	(6.46%)	(40.40%)	(6.21%)	(40.25%)	(6.17%)	(41.77%)	(6.73%)
≥60	1,896,056	160,615	1,947,408	163,694	2,042,630	170,065	2,158,015	182,847	2,069,555	193,376
	(31.10%)	(8.47%)	(32.42%)	(8.41%)	(32.61%)	(8.33%)	(33.90%)	(8.47%)	(35.94%)	(9.34%)
Hospital types *										
Central hospital	1,328,500	106,522	1,332,033	106,470	1,366,387	104,052	1,380,172	105,838	1,281,493	107,002
	(21.90%)	(8.02%)	(22.18%)	(7.99%)	(21.81%)	(7.62%)	(21.68%)	(7.67%)	(22.26%)	(8.35%)
General hospital	1,392,396	86,741	1,406,682	88,244	1,464,010	91,347	1,479,134	93,239	1,363,341	94,977
	(23.00%)	(6.23%)	(23.42%)	(6.27%)	(23.37%)	(6.24%)	(23.23%)	(6.30%)	(23.68%)	(6.97%)
Community hospital	2,627,680	128,040	2,597,254	127,307	2,767,683	132,515	2,843,502	137,393	2,511,310	145,019
	(43.40%)	(4.87%)	(43.24%)	(4.90%)	(44.18%)	(4.79%)	(44.66%)	(4.83%)	(43.61%)	(5.77%)
Private hospital	201,867	9343	175,249	8535	165,714	7947	162,042	7830	132,336	6579
	(3.33%)	(4.63%)	(2.92%)	(4.87%)	(2.65%)	(4.80%)	(2.55%)	(4.83%)	(2.30%)	(4.97%)
Others (e.g., medical university hospital, health center)	506,057	46,699	495,442	44,219	500,867	43,253	501,613	45,287	469,685	48,267
	(8.36%)	(9.23%)	(8.25%)	(8.93%)	(8.00%)	(8.64%)	(7.88%)	(9.03%)	(8.16%)	(10.28%)

* Statistically significant at *p*-value of 0.05.

**Table 3 healthcare-11-01121-t003:** Estimated cost burden of medical harm in each group between 2016 and 2018.

	Year 2016				Year 2017				Year 2018			
	Medical Harm Events (Visit)	Median Cost (USD)	Interquartile Range (USD)	Total Cost (Million USD)	Medical Harm Events (Visit)	Median Cost (USD)	Interquartile Range (USD)	Total Cost (Million USD)	Medical Harm Events (Visit)	Median Cost (USD)	Interquartile Range (USD)	Total Cost (Million USD)
Total visits	424,411	271.74	625.23	261.6	422,869	288.97	623.85	263.58	427,341	299.11	632.85	270.26
Age group *												
0–14	54,269	97.37	169.25	22.53	50,746	104.54	199.79	22.21	51,857	111.19	207.69	22.44
15–59	162,461	248.81	575.26	104.45	160,335	265.40	577.52	103.25	157,192	274.31	583.35	103.07
≥60	160,615	365.34	674.21	134.61	163,694	377.58	663.03	138.12	170,065	379.92	642.20	144.75
Hospital types *												
Central hospital	106,522	430.03	980.47	100.14	106,470	452.22	987.75	99.60	104,052	465.06	1038.32	103.06
General hospital	86,741	290.06	610.50	54.04	88,244	315.35	629.32	57.47	91,347	334.80	652.44	61.38
Community hospital	128,040	163.47	216.12	33.48	127,307	189.55	239.19	36.86	132,515	204.09	253.35	40.57
Private hospital	9343	460.34	1149.85	11.36	8535	443.51	1041.86	10.10	7947	396.09	943.39	8.77
Others (e.g., medical school, health center)	46,699	639.56	1370.86	62.56	44,219	641.64	1399.35	59.55	43,253	613.37	1295.39	56.48

* Statistically significant at *p*-value of 0.05.

**Table 4 healthcare-11-01121-t004:** Estimated cost burden of medical harm in each group between 2019 and 2020.

	Year 2019				Year 2020			
	Medical Harm Events (Visit)	Median Cost (USD)	Interquartile Range (USD)	Total Cost (Million USD)	Medical Harm Events (Visit)	Median Cost (USD)	Interquartile Range (USD)	Total Cost (Million USD)
Total visits	439,386	312.21	645.96	283.63	458,034	343.69	691.14	311.29
Age group *								
0–14	48,726	113.54	231.83	22.23	46,613	122.53	252.35	22.48
15–59	158,014	284.25	602.15	106.28	161,855	316.85	655.73	115.79
≥60	182,847	382.04	644.87	155.12	193,376	420.56	685.59	173.02
Hospital types *								
Central hospital	105,838	487.03	1054.28	106.28	107,002	551.11	1143.58	115.59
General hospital	93,239	342.47	664.62	64.59	94,977	379.76	708.85	72.02
Community hospital	137,393	218.27	257.80	44.40	145,019	247.44	283.71	52.49
Private hospital	7830	441.15	1158.93	9.77	6579	611.97	1389.02	9.51
Others (e.g., medical university hospital, health center)	45,287	631.16	1242.30	58.58	48,267	623.08	1252.33	61.69

* Statistically significant at *p*-value of 0.05.

**Table 5 healthcare-11-01121-t005:** Top 10 medical harm events in inpatient departments, ranked by average prevalence, between 2016 and 2020.

Medical Harm Events (ICD-10 Codes)	Prevalence (Number of Observations or Visits)								
	2016	2017	2018	2019	2020	Average	SD	Median	IQR *
Other sepsis (A41)	106,770	102,618	105,089	112,005	119,242	109,145	6612	106,770	6916
Other specified bacterial agents as the cause of diseases classified to other chapters (B96)	50,798	52,787	53,004	56,985	62,725	55,260	4739	53,004	4198
Decubitus ulcer and pressure area (L89)	39,211	39,587	40,735	42,652	42,011	40,839	1491	40,735	2424
Complications of procedures (T81)	37,000	35,406	33,421	33,025	32,929	34,356	1788	33,421	2381
Other bacterial intestinal infections (A04)	18,039	16,659	16,931	15,495	13,721	16,169	1641	16,659	1436
Perineal laceration during delivery (O70)	18,982	17,625	15,868	14,324	13,819	16,124	2182	15,868	3301
Complications of other internal prosthetic devices, implants and grafts (T85)	11,840	16,197	16,584	17,044	18,414	16,016	2480	16,584	847
Postprocedural respiratory disorders, not elsewhere classified (J95)	14,011	15,180	16,121	16,144	17,362	15,764	1249	16,121	964
Foetus and newborn affected by other complications of labour and delivery (P03)	12,859	12,415	12,170	11,059	12,481	12,197	683	12,415	311
Streptococcus and staphylococcus as the cause of diseases classified to other chapters (B95)	7407	8179	8430	9143	11,154	8863	1424	8430	964

* IQR = interquartile range.

## Data Availability

Data sharing is not applicable to this article.

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
