# Peer review of "How Many People Experience Unsafe Medical Care in Thailand, and How Much Does It Cost under Universal Coverage Scheme?"

_healthcare, 2023, doi:10.3390/healthcare11081121_

Round 1

Reviewer 1 Report

I would like to come in the authors on approaching this very important topic, and looking at a large range of high volume patients over multiple years. This is very important data and help set the stage for future safety and quality measures.
Overall the paper is written soundly and flows quite well. I think there are two areas that can be approved upon that would make it a stronger manuscript.

First, I wonder if statistical analysis could be used to compare some of your outcomes and look for true statistical significance versus just descriptive percentages. A statistician should be able to do this for different things like your age ranges, hospital types, and medical diagnosis which would give us some P values and confidence intervals. That would greatly increase the strength of your presentation; you have very high patient volumes here, so power should not be a problem, and I always believe that statistical analysis is much better than descriptive analysis.

Addressing that problem may help you address my second concern which is the brevity of your discussion. The Discussion in this manuscript is five paragraphs long, and two of those paragraphs and over 60% of the words used in the discussion are all your limitations; that makes your manuscript weaker in a way and less interesting for readers. I think your limitation section is actually written quite well and we keep it how it is as it does address multiple issues which makes sense. Maybe with a statistical analysis as I pointed out in the first portion of my comments, you’ll be able to have a stronger data to discuss further. The discussion could really be strengthened by the authors, pointing out what the impact of their data is, what they think it means, for healthcare in Thailand, and what readers can walk away from in their own countries if they find similar situation’s in their hospital systems. Would love to walk away with a few key points from this discussion. 

Reviewer 2 Report

Comments

The quality of health and patient safety are themes whose concern for various world entities has been intensified. Several documents have already been published with the aim of highlighting the existing problems in terms of quality, the impact of adverse events and the need to invest in patient safety and management of risk with the primary objective of continuous improvement of quality in health. One of the first and most significant references was published by the Institute of Medicine (IOM) through the report “To Err is Human: Building a Safer Health System” in the year 2000, followed by other publications that continued the need to improvement of quality in health (Kohn, Corrigan and Donaldson, 2000). The policy "Patient and Personnel (2P) Safety" falls within the scope of this study.

Administrative routine is not a good adviser to support studies of this nature, although it is important to consider them. Mother data base should be measured.

Infections associated with health care are considered as one of the most adverse events frequent all over the world and a relevant, if not the main, threat to the patient safety (OECD, 2016; US Department of Health and Human Services; WHO, 2016. Its incidence has an impact significant in the quality of care, morbidity, mortality, lengthening hospital stays and worsening health costs (Lambert et al. al., 2011; WHO, 2016).

Sugestions

The data base limits are a considerable obstacle for the implementation of the “Patient and Personnel (“P) Safety” (e.g., costs, hospitals conditions and specialized health professionals). How can this be reversed? The article points to safe results but lacks further complementary scientific analysis (as, by the way, the authors acknowledge). this is a critical point for the elaboration of a Universal Coverage scheme. The article could go further and the time period analysed could be longer than 2016-2020. Missing data?

The methodological component, where the methodology underlying the study, such as the objectives, the study design, the population under study, the criteria of inclusion and exclusion, variables, data sources, data analysis strategy should also address ethical considerations. The diversity of quality analysis approaches and methodologies also depend on the character multidimensionality of the very concept of quality. And finally, the health actions that do not produce goods, which includes immaterial labour.
